# Model Predictive Control: Demand-Orientated, Load-Flexible, Full-Scale Biogas Production

**DOI:** 10.3390/microorganisms10040804

**Published:** 2022-04-12

**Authors:** Celina Dittmer, Benjamin Ohnmacht, Johannes Krümpel, Andreas Lemmer

**Affiliations:** State Institute of Agricultural Engineering and Bioenergy, University of Hohenheim, Garbenstrasse 9, 70599 Stuttgart, Germany; celina.dittmer@uni-hohenheim.de (C.D.); benjamin.ohnmacht@uni-hohenheim.de (B.O.); andreas.lemmer@uni-hohenheim.de (A.L.)

**Keywords:** anaerobic digestion, feeding management, forecast, flexibilization, modeling, demand-driven biogas production

## Abstract

Biogas plants have the great advantage that they produce electricity according to demand and can thus compensate for fluctuating production from weather-dependent sources such as wind power and photovoltaics. A prerequisite for flexible biogas plant operation is a suitable feeding strategy for an adjusted conversion of biomass into biogas. This research work is the first to demonstrate a practical, integrated model predictive control (MPC) for load-flexible, demand-orientated biogas production and the results show promising options for practical application on almost all full-scale biogas plants with no or only minor adjustments to the standardly existing measurement technology. Over an experimental period of 36 days, the biogas production of a full-scale plant was adjusted to the predicted electricity demand of a “real-world laboratory”. Results with a mean absolute percentage error (MAPE) of less than 20% when comparing biogas demand and production were consistently obtained.

## 1. Introduction

Sustainable, low-emission power generation requires a significant expansion of wind power and photovoltaic (PV) plants [1]. Therefore, a transformation of the power grids from a few centralized, controllable fossil or nuclear power plants to several million decentralized wind power and PV plants is needed [2]. Thus, electricity production is increasingly characterized by strong fluctuations that are decoupled from electricity demand.

In Germany, the gross electricity generation from wind power in 2020 was already about 23.5% and from PV about 9% of the total electricity generation [3]. One approach to permanently balance the resulting residual loads and sustainably secure the energy supply is to expand the power grids and increase storage capacities [4]. However, from a technical and economic point of view, these measures can only be implemented to a limited extent [5,6]. Another approach that is becoming increasingly important is the use of controllable producers that offer the advantage of power production driven by demand [7]. With the targets of the German Renewable Energy Act to cover 80% of the gross electricity supply by renewable energies by 2050, energy scenarios predict a need for 60 to 130 GW of controllable power plant capacity to ensure security of supply [8].

“Of the available renewable energy technologies—Apart from geothermal plants and hydropower stations—Bioenergy plants in general and biogas plants in particular, currently enable electricity production to be controlled” [9]. The load-flexible operation of biogas plants could balance both long-term seasonal and short-term fluctuations in electricity feed-in. However, of the approximately 9000 installed biogas-combined heat and power plants (CHP) in Germany, which is equivalent to about 5 GW [10], it is estimated that currently, only 150 biogas plants produce flexible energy [11].

Flexible operation of biogas plants requires technical adjustments, previously limited mainly to the expansion of gas storage capacities [12] and CHP performance [13]. Thus, short-term fluctuations in electricity demand can be balanced by decoupling biogas and electricity production. Nevertheless, the greatest flexibility can only be achieved if a suitable feeding strategy is used for demand-orientated conversion from biomass into biogas [9,12]. Then, biomass serves as a long-term energy storage before it is converted into biogas and further into electricity.

Concepts for feeding strategies were first investigated in laboratory experiments, mostly as batch processes [14,15]. The results obtained in these experiments cannot be adequately transferred to full-scale plants, since these are operated continuously and show entirely different conditions with respect to the population density of microorganisms and the structure of the microbial community, as well as homogeneous substrate availability [16]. Strategies for adapted feeding management at full-scale plants have previously been based on very extensive, complex models. In this context, various studies of models can be found in the literature [17,18,19,20], which primarily aimed at improving process stability and efficiency. Model predictive control (MPC) is particularly suitable for adjusting the biogas plant operation to specified future setpoints [21]. In this process, an internal dynamic model is used to calculate the future behavior of the process as a function of input signals [22]. Models specifically focusing on energy demand-driven biogas production have been developed by Grim, Mauky and Peters et al. [22,23,24], whereas Mauky et al. [21] use MPC to feed the process + to follow a predetermined schedule for gas utilization. All these approaches are based on the well-known Anaerobic Digestion Model No. 1 (ADM1), published by Batstone [25]. The ADM1 is highly complex since it takes into account the parameters of all biochemical process steps (hydrolysis, acidogenesis, acetogenesis, methanogenesis) as wellphysicochemicalical processes (ion association/dissociation and gas-liquid transfer, etc.) [23] which must be determined or estimated individually for each digester and each of the substrates used. Although the complexity of ADM1 has already been significantly reduced by Weinrich and Mauky et al. [21,24,26] , a large number of individual input parameters are still required. These primarily include the composition and degradation behavior, respectively, and the gas formation kinetics of the substrates used. All parameters have to be carefully adjusted to the specific full-scale plant and actual process state [27,28]. In addition, there is a lack of robust measurement technology, that would make the main process states observable, or extensive equipment for process monitoring is usually not available at full-scale plants [29].

The aim of this research is to develop a practical, integrated MPC for load-flexible, demand-orientated biogas production, which can be used on nearly all full-scale biogas plants with no or only minor adjustments to available measurement technology. For this purpose, the application of our previously evaluated forecast model [30] and the simulation model we developed [28] in combination with a new controller is demonstrated here for the first time in full-scale. Here, the biogas production of the full-scale plant is aligned with the predicted power demand of a “real-world laboratory”. The performance is evaluated in terms of adaptability to changing biogas demand profiles.

## 2. Materials and Methods

### 2.1. Experimental Setup

The present study was carried out at the full-scale research biogas plant of the University of Hohenheim, located at “Unterer Lindenhof” in Eningen unter Achalm, Southwest Germany. As described in detail by Dittmer et al. [30], this site functions as a “real-world laboratory” with an agricultural area of almost 180 ha, livestock, laboratory, and technical rooms, and several employees, interns, and apprentices. The energy demand of the “Unterer Lindenhof” is about 750,000 kWh per year and roughly corresponds to that of a small village with 150 inhabitants.

The biogas plant consists of two continuous stirred-tank reactors and a secondary digester, each with a networking volume of 850 m^3^. Both digesters are covered with insulated concrete and are operated in the mesophilic range at 43 ± 4 °C. Two double-membrane gas storages of 300 m^3^ and 1600 m^3^, installed on the secondary digester and the digestate storage, respectively, serve as buffers for the produced biogas prior to its utilization by a CHP-unit with an installed electrical power of 355 kW_._ A more detailed description of the setup of the entire research biogas plant can be found in Naegele et al. [31,32].

For this study, one of the two primary digesters was used. The digester is equipped with a submersible motor mixer and an inclined agitator unit. The mixing time during the experiment was set so that both mixing devices were operated only two minutes before and after feeding, as well as during feeding. Solid substrates were supplied by a vertical mixer feeding system with a capacity of about 20 tons (Biogas Höre GmbH, Orsingen-Nenzingen, Germany, type Höre Combi 4500). Weighing cells recorded the quantity of substrates used. Liquid substrates were added with a progressive cavity pump and measured with a flow meter (Proline Promag 50; Endress+Hauser, Weil am Rhein, Germany). The volumetric gas flow was detected directly at the gas outlet of the digester by a flow meter with an oscillating measuring method (hot wire sensor, company Esters Elektronik GmbH, type GD300). Temperature and pressure were recorded to calculate the biogas volume produced under standard conditions (1013 mbar, 273 K), which are continuously available in m^3^ h^−1^.

The research biogas plant is equipped with a central plant control (CPC) unit for data collection, storage, and evaluation. Data from the CPC unit is transferred to a relational database for further consolidation, transfer, and processing. Individual programs such as the forecast modules and feed-manager query the database and upload the results of the calculations back for further analysis. Exported feeding schedules from the database can then be loaded into the CPC unit. 

All programming for visualization, modeling, and simulation was done using the programming language R [33]. Specific packages are referenced below wherever appropriate.

### 2.2. Sampling and Analysis

During the experimental period, samples of all substrates used and the digestate were taken three times a week (Mondays, Wednesdays, Fridays) and analyzed in the biogas laboratory of the University of Hohenheim. Samplings were carried out in accordance with the guideline 4630 of the Society for Energy and Environment, documented in [34].

The substrates used and the samples from the digester were analyzed for their dry matter (DM) and organic dry matter (oDM) content. The procedure for this analysis was carried out according to standard methods and is described in detail in [35]. Further, the samples from the digester were analyzed for buffer capacity (FOS/TAC), which was carried out according to the VDLUFA standard methods described in [36]. In addition, the intermediates were investigated by determining the content of acetic, propionic, butyric, valeric, and caproic acids using a gas chromatograph (GC, type CP3800 with flame ionization detector, WCOT Fused Silica capillary column, Agilent Technologies Germany GmbH, Böblingen, Germany).

The results were used to determine major deviations from standard conditions and to calculate the organic loading rate (OLR), which refers to the amount of organic material per unit reactor volume per day.

### 2.3. Model Predictive Control

As illustrated in the schematic diagram in Figure 1, the MPC for load-flexible, demand-orientated biogas production is based on three key components: Forecast of the future power demand for 48 h of the “real-world laboratory”.Forecast of the biogas demand, derived from the power demand forecast.Feeding management model for planning the corresponding substrate-supply.

#### 2.3.1. Power Demand Forecast

The forecasting model developed by Dittmer et al. [30] was used to predict the power demand of the “real-world laboratory”. For this purpose, the historical power consumption data of the last four weeks was retrieved daily to forecast the next 48 h. The model is based on a time series analysis, using correlations of past values, trends, and seasonality to predict future values.

#### 2.3.2. Biogas Demand Forecast

Using fixed conversion factors, the resulting time series of biogas demand was determined from the power demand according to Equation (1). It is the declaration of the required raw biogas with approx. 60% methane content. In addition, a scaling factor and the number of digesters were applied to scale the “real-world laboratory” to the size of the actual CHP unit and to simulate the adjustment of the biogas demand to larger or smaller consumption units to test different settings of the model.
(1)$$D_{t}^{s}=\left(\frac{\frac{P_{fcst}\;\cdot f_{s}}{\eta_{BHKW}}}{f_{\mathrm{kWh}\_m^{3}}}\right)/n\;\;$$

The parameters are defined as:

$D_{t}^{s}$ biogas demand in m^3^ with index *t* for the time and *s* stands for the series*P_fcst_* forecasted power demand in kWh*f_s_* scaling factor *η_BHKW_* fixed efficiency of CHP of 0.35*f*_kWh_m^3^_ fixed conversion factor from kWh to m^3^ -of 5.5 kWh per m^3^*n* number of digesters

#### 2.3.3. Feeding Management Model

The workflow of the feeding management model is as follows:

Random construction of feeding timetables for the next 48 h. Simulation of biogas production for each feeding timetable.Selection of the timetable fitting best the demand.

#### 2.3.4. Randomly Constructed Feeding-Timetables (1.)

Considering the technical limitations of the full-scale biogas plant and to simplify the process in general, only feeding increments of 500 kg were allowed in this experiment. The 48-h horizon was divided into 5-min intervals. For each interval, a binomial distribution was generated (0 or 1), corresponding to a feeding event of 500 kg or no feeding. Additionally, a “Probability of success” (POS) was introduced to determine the likelihood of the outcome to feed. The POS was calculated each day based on the total biogas demand in the next 48 h and the historical specific biogas-yield of the last three weeks (see Equations (2) and (3)).
(2)$$Specific\;biogas\;yield=\frac{\bar{b_{hist}}}{\bar{s_{hist}}}\;$$
(3)$$\mathrm{POS}=\frac{\sum D_{t}^{s}}{Specific\;biogas\;yield\;\cdot s_{fixed}}/\;576\;$$

The parameters are defined as: 

$\bar{b_{hist}}$ mean biogas production over the last 500 h in m^3^ h^−1^$\bar{s_{hist}}$ mean substrate feeding over the last 500 h in kg h^−1^$\sum D_{t}^{s}$ total biogas demand over the next 48 h in m^3^$s_{fixed}$ fixed substrate quantity of 500 kg576 related to the number of 5-min intervals during 48 h

The POS is used to reduce the total number of random timetables, or in other words, it creates a set of feeding timetables that meet the requirements of the total biogas needed in the next 48 h, rather than creating entirely random timetables. 

Finally, the 5-min intervals with their corresponding binomial values are aggregated into an hourly time series.

In total, 1500 timetables are created for the following 48 h, on each day of the experiment.

#### 2.3.5. Simulation of Biogas Production (2.)

For each of the 1500 randomly constructed feeding plans, the resulting biogas production is simulated. For this, the novel simulation model developed by Dittmer et al. [28] is used. The model is based on correlations between substrate supply and biogas production, which are determined over a training horizon of the last 500 h of observation. Including the recent feedings and the new random-timetables, the resulting biogas production can be simulated by Equation (4) [35]:(4)$$Y_{t}=\alpha +\beta_{0}X_{t}+\beta_{1}X_{t-1}+\ldots +\beta_{k}X_{t-k}+\varepsilon_{t}\;$$

The parameters of the multiple regression model are defined as: 

Y dependent variable*t* point in timeX independent variableα interceptβ slope$\varepsilon_{t}$ error term*k* lag order

#### 2.3.6. Select the Best Feeding-Timetable (3.)

The appropriate feeding plan is selected by comparing the simulated biogas production to the biogas demand, using the symmetric mean absolute percentage error (*SMAPE*) [37]. The SMAPE is defined as follows:(5)$$\mathrm{SMAPE}=\frac{\left|D_{t}^{s}-z_{t}^{s}\right|}{\left(\left|D_{t}^{s}\right|+\left|z_{t}^{s}\right|\right)}\;$$

The parameters are defined as:

$z_{t}^{s}$ simulated values*t* point in time*s* for series$D_{t}^{s}$ is defined in Equation (1)

The feeding-timetable with the lowest SMAPE is selected, uploaded to the database, and read into the plant-control to be executed.

### 2.4. Practical Implementation, Experimental Phases

The daily imported feeding plan contains the specification of the time of the day when feeding is to take place and the respective quantity of solid substrates to be supplied into the digester. To ensure proper reading and processing of all required data, a time window of three hours was specified during which no feedings were permitted in the feeding plan. This time window was also used to fill the vertical mixer feeding system with sufficient solid substrates to ensure that at least about 12 tons were in stock. This ensured that there was always enough material for all feedings over the next 24 h. With this exception of the daily supply of the feeding system, all process steps of the experiment ran completely automatically.

The experiment was divided into four phases in which different settings of the model were created and tested. Prior to the experimental phases, a 29-day start-up phase was conducted to optimize the integration of the model into the CPC and the automation of the process steps. Furthermore, stable digestion conditions were ensured by adjusting the feeding quantities as required in the initial experimental phase and, accordingly, maintaining an approximately constant OLR. Data collected during this phase were not included in the evaluation of the model. 

During the four experimental phases, the biogas demand was scaled alternately to half and back to full demand, to test the suitability and limitations of the model with regard to major fluctuations in electricity demand. This was done by setting the scaling factor $f_{s}$ to the value 4 or 2 in Equation (6).
(6)$$D_{t}^{s}=\left(\frac{\frac{P_{fcst}\;\cdot f_{s}}{\eta_{BHKW}}}{f_{\mathrm{kWh}\_m^{3}}}\right)/n\;\;$$

Accordingly, the first phase was conducted with a high demand and covered 360 h/15 days. In the second phase, the demand was reduced by over 216 h/9 days. Subsequently, in a third phase, the demand was increased again over 168 h/7 days and finally halved again over 120 h/5 days (fourth phase).

### 2.5. Evaluation of the MPC

MPC was evaluated using various accuracy parameters. These were used to assess the average quality of the results per day. In addition to evaluating the overall model (biogas demand versus production), the substeps, such as simulation versus demand and simulation versus actual biogas production, were also assessed.

First, the SMAPE was calculated, since it is also used as a selection criterion in the model itself. The calculation can be found in Equation (7). Although other parameters are used in this context, for which reason the index *Q* is used:(7)$${\mathrm{SMAPE}}_{Q}=\frac{1}{E}\overset{E}{\underset{t=1}{\sum }}\frac{\left|actual-predicted\right|}{\left(\left|actual\right|+\left|predicted\right|\right)}\cdot 100\%\;$$

The parameters are defined as:*actual* set point in m^3^ h^−1^*predicted* modeled value in m^3^ h^−1^

In Equation (7), E indicates the number of observations, which here always includes 24, since the parameter is considered per day.

Furthermore, the commonly used mean absolute percentage error (MAPE) was applied [37]. The parameters and abbreviations are the same as in Equation (8).
(8)$$\mathrm{MAPE}=\frac{1}{E}\overset{E}{\underset{t=1}{\sum }}\left|\frac{actual-predicted}{actual}\right|\cdot 100\%\;$$

In addition, the mean absolute error (MAE) and the positive and negative deviations from the actual and predicted values of the overall model (biogas demand versus production) were used as parameters for the evaluation. For this purpose, the R function *mae()* from the R package *metrics* [37] and differences between actual and predicted values were calculated. These daily deviations are particularly relevant for the evaluation of the compensation of over- or underproduction by the gas storage.

## 3. Results and Discussion

### 3.1. Power Demand Forecast and Derived Biogas Demand

Derived from the projected electricity demand, Figure 2 shows the hourly biogas demand over the entire period of the experiment.

Variations resulting from day/night and weekday/weekend cycles were observed, which is typical for the energy demand of a site, as described in Dittmer et al. [30]. At our site, demand during the day largely exceeds demand during the night, and less energy is consumed during weekends than on weekdays. The largest variation in biogas demand within one hour was observed in the transition from one phase to the other, with differences of nearly 45 m^3^ h^−1^ in each case. This leap is relatively high compared to the results of Mauky et al. [21] with, for example, a difference of 20 m^3^ h^−1^ only. The largest spread within one day was 95 m^3^ on day 25.

The mean biogas demand in the first experimental phase was 107 m^3^ h^−1^, with a maximum of 170 m^3^ h^−1^ and a minimum of 40 m^3^ h^−1^. This first experimental phase ended after day 15 and represented the longest phase. Since this phase covered two whole weeks, the different demands of weekdays and weekends were sufficiently represented.

The second experimental phase shows the biogas demand over nine days with a weekend in the middle. The mean demand during this phase was 51 m^3^ h^−1^, while maximum and minimum were 95 m^3^ h^−1^ and 36 m^3^ h^−1^, respectively. Thus, the biogas demand was reduced by about 50% compared to the first experimental phase. This reduction in demand should demonstrate how reliably and quickly biogas production can be significantly reduced.

In the third experimental phase, the biogas demand was increased again, with a mean demand of 100 m^3^ h^−1^ during a period of seven days and maximum and minimum values of 146 m^3^ h^−1^ and 47 m^3^ h^−1^, respectively. This phase was intended to show the extent to which a rapid significant increase in biogas production is possible.

For reproducibility, the last phase over five days represents again a reduced demand with a mean demand of 49 m^3^ h^−1^ and a maximum and minimum of 68 m^3^ h^−1^ and 37 m^3^ h^−1^, respectively.

### 3.2. Feeding Management Model

#### 3.2.1. Calculation Results

36 individual feeding schedules with a 48 h time horizon were calculated for each day of the entire experimental period and are shown in Figure 3 with the corresponding biogas simulations. 

The lower part of Figure 3 shows the planned feeding schedules. Remarkably, even in the phases with higher biogas demand, 500 kg of substrate per hour should be used in most cases, but 1000–3000 kg in rare cases. However, based on the boundary conditions of the model, a maximum quantity of 6000 kg h^−1^ would be possible. This maximum feeding rate was chosen due to technical limitations of the biogas plant (solid feeding system, stirring technology, etc.) and the OLR. In this respect, adjustments in the model might be necessary when it is adapted to other biogas plants.

The simulated biogas production, especially in the first experimental phase, closely followed the biogas demand with only a few deviations. The fact that especially the first phase provides comparatively good results is also demonstrated by the accuracy parameters in Table 1, with a mean daily MAPE of 7.3%. This can be explained, among other things, by the fact that the biogas plant was already operated with similar conditions in terms of average biogas demand, OLR, and substrate mix in the 29-day start-up phase. This allowed the simulation model, which uses historical data as shown in Equation (4), to adapt to the conditions [28].

The third phase, which also represents high biogas demand, yielded good results as well. Here, the average daily MAPE was 9.6%. In particular, the transitions to an experimental phase with lower demand were characterized by larger deviations. This suggests that the decrease in biogas production is slower to realize than the increase, which is a result of the slow degradability of the substrates. However, this is taken into account by the simulation model [28]. Overall, the largest deviations were observed in phase two, with an average MAPE of 25%. However, after day 4 of experimental phase two, the deviations decrease significantly. Phase 4, in turn, shows better values with an average MAPE of 15%.

A major reason for the relatively poor simulation results in phase two is the use of historical biogas production data by the model. As described in Dittmer et al. [28], the results of the simulations are based on Equation 4 with the data of the biogas production and the feeding amounts of the last 500 h. Since the biogas production during this period was never as low as in phase two, the model has to level off first. In this case, the model will first output the value of the intercept α. This will only reach the required low values when the historical data on which the calculation is based also include lower values. Therefore, the situation is different in phase four, where training data of a similarly low demand of phase two are already included and thus, accurate simulation results could be obtained much faster.

#### 3.2.2. Implementation Results

The results of the implemented procedure for the feeding management are presented in Figure 4 in red for the produced biogas and the actual substrate feeding.

During the first experimental phase, the biogas produced closely matched with the simulated data and the biogas demand profile, which is also confirmed by the accuracy parameters in Table 2, with a mean daily MAPE of the simulation results of 10.2%. The only major deviation around the 9th and 10th day of the experiment can be explained by planned but not executed feedings, as can be seen in the lower part of Figure 4. These feedings were not executed because of technical problems in the control system of the biogas plant. The same problem was also observed in phase three around days 29 and 30, resulting in a mean MAPE in this phase of 18.8%.

The experimental phases two and four with low biogas demand showed a mean MAPE value of 26.3% and 11.8%, respectively. In particular, the second phase showed slightly poorer results. However, this can be justified, by the fact that the simulation results already deviated clearly from the biogas demand profile, as discussed in the previous section. On the other hand, in line with the regular operation of the full-scale biogas plant and thus, regardless of the feeding management, daily feedings of 6000 kg of liquid manure were carried out during the experiment. These resulted in an increase in biogas production that could not be accounted for by the model, since liquid manure feeding is not considered as a parameter [28]. Consequently, the liquid manure supply was distributed more evenly throughout the day, resulting in better simulation results from day 21 onwards, as shown in Figure 4 and confirmed by the accuracy parameters in Table 2.

In summary, biogas production could be simulated sufficiently well with an average daily MAPE of 16.1% over the entire experimental period. These results are consistent with the findings of Dittmer et al. [28], who reported an average MAPE of 14–18% when evaluating 366 simulations.

As can be seen in Figure 4, feedings could be carried out as planned, with the exceptions mentioned above. In order to evaluate the present model in more detail, further experiments with a more variable substrate composition might be useful. Solid substrates were used in the composition of 41% solid manure, 52% corn silage, and 7% grass silage on average during the experimental period. The liquid manure input, which is not taken into account in the model, also accounts for 58% of the total substrate mass used. This represents more than 50% of the total feed. This again justifies the significant increase in biogas production when the liquid manure was batched and without simultaneous feeding of solid substrates. However, due to the relatively low biogas yield [38] and the results of Dittmer et al. [28], it is justified not to include liquid manure feeding as an additional parameter in the simulation model.

To evaluate the stability of the process conditions during the experimental period, the OLR was calculated, which was on average 1.65 kg dry organic matter per cubic meter of digester volume per day. This value tends to be low with regard to the potential use of a biogas plant of this size [39]. This can be explained by the fact that the biogas demand profile was generally set low during the experimental period and was also reduced twice by 50% in each case. Therefore, high dosages were also rarely planned in the feeding schedule and the maximum of 6000 kg per hour was never provided. The results suggest that, with regard to OLR, a larger biogas demand could also be met by higher or more frequent substrate dosages.

Considering the quality of the final results of the feeding management model, balancing biogas demand and production resulted in an average daily MAPE of 17.6% over the entire experimental period. Phase one showed the best results with a low MAPE of 11.6% on average, while the latter three phases showed average MAPE values between 20–23%. The actual difference between biogas demand and production was calculated as a further quality parameter. Here, it was assumed that the biogas demand, which was derived from the forecasted electricity demand, actually occurred in this way (no forecast errors). Positive values indicate overproduction, whereas negative values indicate insufficient biogas production. On day 30 of the experimental period, the lowest negative value was recorded with a difference of 714.9 m^3^. However, this resulted from the technical problems already explained above and not from the unsatisfactory results of the model. The second highest deviation of −640.1 m^3^ was calculated for day 25. Maximum overproduction was reached on day 32 with 486.1 m^3^. On average over the entire experimental period, 92.1 m^3^ of biogas per day were underproduced.

In particular, the consideration of the over- and underproduction of biogas during the experiment serves as an estimate for the required gas storage capacities in the case of demand-oriented biogas production using the model presented. Based on the research for a full-scale biogas plant, the total gas storage volume is 950 m^3^ per digester (two double membrane gas storages with 300 m^3^ and 1600 m^3^ capacity are available for both digesters). Accordingly, both under- and overproduction could be compensated in this case. However, this assumes an almost empty gas storage at the same time of maximum overproduction and vice versa. This is a crucial factor since the upgrading of the gas storage capacity is technically limited and costly, as discussed in [9,40,41]. In this context, a potential improvement of the present model would be to include the gas storage level in the calculation of the predicted biogas demand by adjusting it downwards or upwards accordingly. This is relatively simple and could possibly lead to a further reduction of the required gas storage capacity.

Moreover, further research could aim to integrate CHP-control into the MPC to represent the complete full-scale biogas plant. For example, this would allow biogas production to be based not only on electricity demand, but also on the thermal demand of a site. This would include an additional algorithm for the continuous calculation of adjusted CHP schedules. Furthermore, large-scale experiments of the MPC on several full-scale biogas plants would be useful to further verify the present results. Also, a much longer experimental period would be beneficial, especially in view of the changing substrates used and large fluctuations in the biogas demand profile.

## 4. Conclusions

The future operation of biogas plants should be demand-orientated Therefore, a new MPC is presented, which is highly practicable for the first time and consistently yields results with a MAPE of less than 20% comparing biogas demand and production. Over an experimental period of 36 days, the biogas production of a full-scale plant was adjusted to the actual electricity demand of a “real-world laboratory”, which was predicted from the demand during 48 h in advance. The calculation of the feeding plans used a type of Monte Carlo algorithm followed by a simulation of the expected biogas production. The most suitable feeding plan contains only the information about the amount of solid substrates to be used and the respective time of feeding. Thus, the presented MPC differs significantly from all previous models, since it requires very few input parameters. Specifically, only the measured actual electricity demand (using regularly installed electricity meters), the quantity of solid substrates fed in, regardless of their composition, (e.g., weighing cells under the solids feeder), and for the biogas produced (e.g., flow meters) need to be available. Therefore, the results show promising options for practical application nearly all full-scale biogas plants with no or only minor adjustments of the technical setup.

## Figures and Tables

**Figure 1 microorganisms-10-00804-f001:**
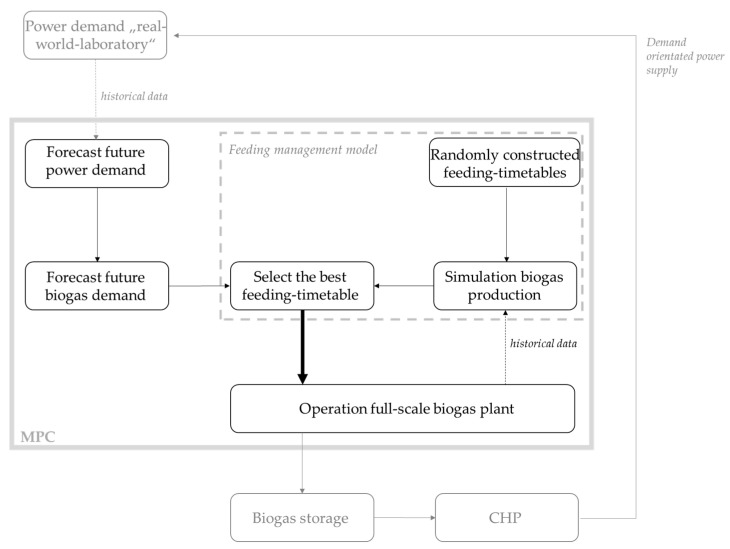
Schematic representation of the model predictive control (MPC) with showing and classifying the three key components: forecast of future power demand, forecast of biogas demand, feeding management model.

**Figure 2 microorganisms-10-00804-f002:**
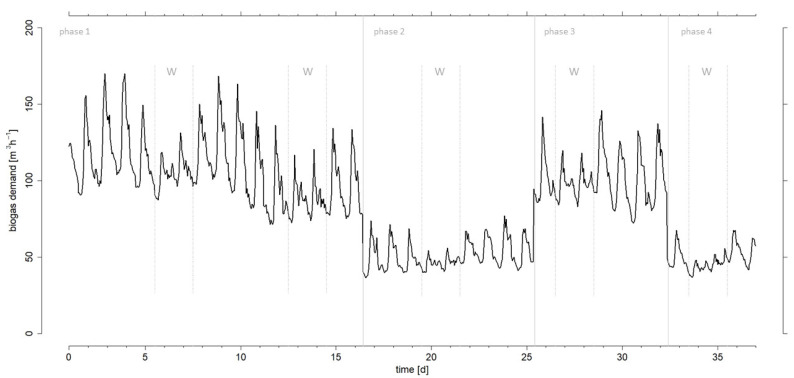
Hourly biogas demand profile over the entire test period, derived from the electricity demand of the “real-world laboratory”. The vertical grey lines identify start and end of the four experimental phases. The weekends are marked with dotted lines and “w”.

**Figure 3 microorganisms-10-00804-f003:**
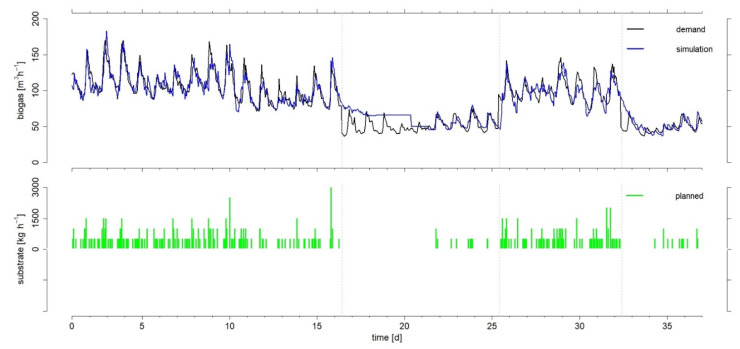
Biogas demand and simulated biogas production, as well as the associated daily feeding schedules in hourly resolution over the entire experimental period. The vertical grey lines identify start and end of the four experimental phases.

**Figure 4 microorganisms-10-00804-f004:**
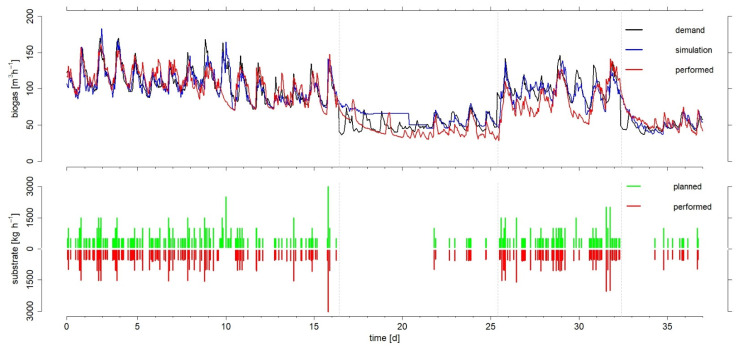
Biogas demand, simulated biogas production and biogas produced, as well as the associated daily feeding schedules and the feedings performed in hourly resolution over the entire experimental period.

**Table 1 microorganisms-10-00804-t001:** Mean values of accuracy parameters SMAPE [%], MAPE [%], and MAE [m^3^] to evaluate the quality of simulation results.

	Phase 1	Phase 2	Phase 3	Phase 4
Time Period	Day 1–15	Day 16–24	Day 25–31	Day 32–36
SMAPE [%]	7.4 ± 1.7	20.3 ± 11.6	10.1 ± 2.5	12.5 ± 6.0
MAPE [%]	7.3 ± 1.7	24.9 ± 16.0	9.6 ± 2.0	14.5 ± 9.2
MAE [m^3^]	8.1 ± 2.1	11.6 ± 6.9	9.6 ± 1.5	7.0 ± 4.5

**Table 2 microorganisms-10-00804-t002:** Mean values of accuracy parameters SMAPE [%], MAPE [%] and MAE [m^3^] as well as the difference between biogas demand and production (diff. [m^3^]) for evaluating the quality of simulation (comparison data of the simulation and the biogas produced) [Sim] and of the final results of the whole feeding management model (comparison data biogas demand and the biogas produced) [Feed].

	Phase 1	Phase 2	Phase 3	Phase 4
Time Period	Day 1–15	Day 16–24	Day 25–31	Day 32–36
SMAPE [%]				
[Sim]	10.3 ± 4.2	31.3 ± 10.3	21.6 ± 8.3	11.3 ± 4.0
[Feed]	11.7 ± 4.6	24.0 ± 3.8	25.1 ± 9.0	17.2 ± 6.4
MAPE [%]				
[Sim]	10.2 ± 3.2	26.3 ± 7.4	18.8 ± 6.4	11.7 ± 4.3
[Feed]	11.6 ± 3.9	23.0 ± 5.3	21.5 ± 6.3	19.9 ± 9.9
MAE [m^3^]				
[Sim]	11.2 ± 4.7	16.1 ± 4.9	17.4 ± 5.6	6.4 ± 1.7
[Feed]	12.5 ± 4.6	11.7 ± 2.4	20.7 ± 5.2	9.8 ± 5.3
diff. [m^3^]				
[Feed]	16.8 ± 212.7	−147.12 ± 195.4	−432.21 ± 277.9	156.14 ± 203.7

## Data Availability

Not applicable.

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
