# Peer review of "Model Predictive Control: Demand-Orientated, Load-Flexible, Full-Scale Biogas Production"

_microorganisms, 2022, doi:10.3390/microorganisms10040804_

Round 1

Reviewer 1 Report

This study mainly focused on the development and validation of a new model predictive control tool for future biogas plant operation. Basically, this is an interesting study; however, the manuscript in the current version has lots of issues. Specific comments and suggestions are provided for a suggested major revision.

  1. To validate the efficiency of the proposed model, the authors only used an experimental period of 36 days, which seems too short to represent a practical anaerobic digestion performance. Sometimes, it takes many days to start up and allow the microbial communities to adapt to the feedstock. Thus, the reviewer thought the data for model validation is not sufficient. That could be part of reasons for the relatively high “mean absolute percentage error (MAPE) of less than 20 %” in the current study.

  1. Line 82: removing one word from “research study”

  1. The authors are suggested to provide a schematic figure for Experimental Setup in Section 2.1.

  1. In the current manuscript, there are lots of “Figure Error! Reference source not 289 found.” and “Table Error! 341 Reference source not found.”. These errors made the reader hard to follow.

Line 289;

Line 331;

Line 338;

Line 341-342;

Line 372-373;

Line 379;

Line 384-385;

Line 404

  1. How can the proposed model be used easily for a fixed biogas plant? The authors are suggested to provide some instructions for practical application of their model in the last paragraph of the manuscript.

Reviewer 2 Report

The manuscript microorganisms-1631092 deals with the demand-oriented modeling and control of feedstock loading in a full-scale biogas plant. The work is presented in a clear way and of scientific and practical interest. The results and discussion are sound. It is worth publication in Microorganisms, in my opinion, with a major revision addressing my comments and questions as follows.

Please find my major comments on this work:

  1. Lines 253-255, what kind of scenarii (different reality) do the four phases of biogas demand represent respectively?
  2. As mentioned in Lines 407-408, so a fixed composition of the solid substrates was imposed? As a result, the biogas demand forecast is only dependent on the quantity of solid feedstock introduced, no more on its seasonal variation, composition and quality? Would it be possible to take these factors into consideration by integrating a simple biochemical model?
  3. Lines 392-394 and Lines 408-414, I do not understand well the discussion about the liquid manure input and its role. It serves as the inoculum of the digestion?
  4. The whole paper is talking about the biogas demand. What is the composition of biogas here? Biomethane (>95%) after purification? Or a raw biogas with a lower biomethane content that allows to feed a generator? Please give more information to clarify this issue.
  5. Lines 458-464, the cogeneration for heat and power production is the unique method to valorizing biogas. What would the modeling strategy change if other valorization methods were considered (like injection to the gas network, feeding vehicles etc.)? It would be interesting to discuss about it to enhance the significance of the work.
  6. Line 193, what does “576” mean in the Equation 3? By referring to my question 2, the specific biogas yield is based only on biogas production during previous 500 hours (21 d), not on the kinetic aspect of substrates biogas production itself. What are the possible advantages and disadvantages of this consideration? Please further comment.
  7. Equation 3, what if more than one feeding plan among 1500 found as best solutions for feeding control?
  8. Section 2.3, a scheme type "flowchart" describing the entire demand-oriented model control would be helpful tofor readers’ understanding of models.
  9. Lines 355-356 and Figure 3, during Phase 2 with a sudden drop of biogas demand, is it really necessary to reduce the biogas production by cutting the substrate feed? Since the biogas could be stored elsewhere, the overproduction of biogas with regard to the demand does not necessarily lead to the use of all biogas produced. We could adjust the performance of cogeneration, instead, in order to satisfy the targeted power demand. The stop followed by a sudden increase of substrate feeding may disturb the anaerobic digestion process and cause dysfunction of the system even though this was not seen in Phase 3. Secondly, the degradation of organic matter in substrate continues to take place during storage, which means if we do not inject the fresh substrate into reactors in time, we lose the methane potential during the low feeding period of Phase 2. Please comment.

Some minor points on the typesetting:

  1. All of the figures cited in the text show an error.
  2. Line 113, und => and ?
  3. Repetition of equation 1 (Line 161 and 166) and equation 4 (Line 214 and Line 217), equation 5 (Line 229 and Line 231) and equation 6 (Line 270 and line 272).
  4. Line 170, please specify unit of Dst.

            According to the comments above, I suggest the major revision of the manuscript.

Round 2

Reviewer 1 Report

The authors have revised the manuscript and it is now acceptable.

Author Response

Dear reviewer,

thank you again for your very helpful review comments. 

Kind regards,

Celina Dittmer

Reviewer 2 Report

The author's comments and the corresponding revision are sound. I suggest accept the paper.

Author Response

(The authors gave the same response as above.)
